# Surrogate Biomarker Prediction from Whole-Slide Images for Evaluating Overall Survival in Lung Adenocarcinoma

**DOI:** 10.3390/diagnostics14050462

**Published:** 2024-02-20

**Authors:** Pierre Murchan, Anne-Marie Baird, Pilib Ó Broin, Orla Sheils, Stephen P. Finn

**Affiliations:** 1Department of Histopathology and Morbid Anatomy, Trinity Translational Medicine Institute, Trinity College Dublin, D08 W9RT Dublin, Ireland; murchanp@tcd.ie (P.M.); osheils@tcd.ie (O.S.); 2The SFI Centre for Research Training in Genomics Data Science, University of Galway, H91 CF50 Galway, Ireland; 3Trinity St. James’s Cancer Institute (TSJCI), St. James’s Hospital, D08 RX0X Dublin, Ireland; bairda@tcd.ie; 4School of Medicine, Trinity Translational Medicine Institute, Trinity College Dublin, D02 A440 Dublin, Ireland; 5School of Mathematical & Statistical Sciences, University of Galway, H91 TK33 Galway, Ireland; pilib.obroin@universityofgalway.ie; 6Department of Histopathology, St. James’s Hospital, James’s Street, D08 X4RX Dublin, Ireland

**Keywords:** deep learning, computational pathology, biomarkers, survival, lung adenocarcinoma

## Abstract

Background: Recent advances in computational pathology have shown potential in predicting biomarkers from haematoxylin and eosin (H&E) whole-slide images (WSI). However, predicting the outcome directly from WSIs remains a substantial challenge. In this study, we aimed to investigate how gene expression, predicted from WSIs, could be used to evaluate overall survival (OS) in patients with lung adenocarcinoma (LUAD). Methods: Differentially expressed genes (DEGs) were identified from The Cancer Genome Atlas (TCGA)-LUAD cohort. Cox regression analysis was performed on DEGs to identify the gene prognostics of OS. Attention-based multiple instance learning (AMIL) models were trained to predict the expression of identified prognostic genes from WSIs using the TCGA-LUAD dataset. Models were externally validated in the Clinical Proteomic Tumour Analysis Consortium (CPTAC)-LUAD dataset. The prognostic value of predicted gene expression values was then compared to the true gene expression measurements. Results: The expression of 239 prognostic genes could be predicted in TCGA-LUAD with cross-validated Pearson’s R > 0.4. Predicted gene expression demonstrated prognostic performance, attaining a cross-validated concordance index of up to 0.615 in TCGA-LUAD through Cox regression. In total, 36 genes had predicted expression in the external validation cohort that was prognostic of OS. Conclusions: Gene expression predicted from WSIs is an effective method of evaluating OS in patients with LUAD. These results may open up new avenues of cost- and time-efficient prognosis assessment in LUAD treatment.

## 1. Introduction

Lung cancer remains one of the most commonly diagnosed cancers globally and is the leading cause of cancer-related mortality [1]. Lung adenocarcinoma (LUAD) is particularly prevalent representing between 50 and 60% of non-small cell lung cancers [2]. As personalised medicine becomes more integrated into clinical practice, prognostic and predictive biomarkers have become indispensable tools for treatment stratification and decision-making [3]. However, reliance on invasive tissue biopsies and costly genomic assays poses significant challenges globally to the broader integration of cancer biomarkers in the diagnostic pathway. Resource limitations, prolonged processing times, and patient health status often restrict access to recent advancements in cancer treatment. This issue is compounded as the number of recognised cancer biomarkers expands with ongoing research.

Computational pathology aims to make use of readily available data to open new avenues for cost-effective biomarker evaluation. In particular, machine learning applied to haematoxylin and eosin (H&E) stained whole-slide images (WSIs) has shown promise in predicting various molecular biomarkers [4,5,6,7]. However, end-to-end strategies, in which machine learning is used to directly predict clinical outcomes, remain a challenge [8]. Multiple studies have investigated the ability of deep learning methods to directly predict genomic and tumour microenvironment features directly from WSIs [9,10,11,12]. For example, Echle et al. previously demonstrated that microsatellite instability can be predicted from WSIs with clinical-grade performance in colorectal tumours, while Schmauch et al. introduced the HE2RNA model to predict bulk RNA-seq profiles from WSIs [9,13]. More recently, Alsaafin et al. showed that a deep learning model trained to learn gene expression patterns could be used for WSI search [14]. However, the prognostic insights that can be drawn from predicting genomic features such as gene expression remains to be explored.

Extensive research has been dedicated to identifying prognostic gene expression signatures from transcriptomic data in LUAD. Various methods such as Cox regression, random survival forests, and deep neural networks have been implemented by researchers with the aim of stratifying patients into low- and high-risk groups [15,16,17,18,19,20]. Researchers have also explored the use of multimodal artificial intelligence (AI) models aimed at evaluating survival, integrating clinical, genomic, and histopathological information [21,22]. However, comprehensive genomic profiling of tumours is not routine in clinical practice, and therefore, it may be several years before patients benefit from such studies. Meanwhile, gold-standard histopathological assessment carried out by a trained pathologist is widely regarded as routine and mandatory in most cancers, resulting in a trove of readily available information in the form of H&E-stained glass slides which are increasingly becoming digitised in clinical practice and in clinical trials. Applying AI to these data will undoubtedly open up new avenues for cost- and time-efficient diagnostics.

In this study, we investigated the clinical value of surrogate biomarkers, specifically gene expression for overall survival (OS) as predicted directly from WSIs. We evaluated the predictability of gene expression markers directly from WSIs using deep learning and assessed the prognostic value of the predicted gene expression.

## 2. Materials and Methods

### 2.1. Data

Data for the TCGA-LUAD data set (*n* = 589) was obtained from the Genomics Data Commons (GDC) data repository (https://portal.gdc.cancer.gov/; accessed on 25 September 2023). Gene expression data was acquired using the *GDCquery* function from the *TCGABiolinks* R package with *data.category* and *workflow.type* set to ’Transcriptome profiling’ and ’STAR counts’, respectively [23]. “Unstranded” gene expression counts were used for the analysis. Diagnostic whole-slide images (WSIs) of formalin-fixed paraffin-embedded tumour tissue from TCGA-LUAD were used to train models. The CPTAC-LUAD dataset was used for external validation of models trained to predict gene expression [24]. CPTAC gene expression data were downloaded from the GDC data repository, while clinical data were downloaded from LinkedOmics (https://linkedomics.org/login.php#dataSource; accessed on 25 September 2023). WSIs for the CPTAC-LUAD cohort were retrieved from The Cancer Imaging Archive (https://www.cancerimagingarchive.net/; accessed on 2 October 2023). Differences in clinicopathological characteristics between the TCGA-LUAD and CPTAC-LUAD cohorts were evaluated using Fisher’s exact test, the Chi-square test, and the Mann–Whitney U test. Statistical tests were implemented using the Python *scipy* library (version 1.10.1) [25].

### 2.2. Differential Expression Analysis

Differential expression analysis was performed between tumour (*n* = 530) and normal (*n* = 59) samples from the TCGA-LUAD using the *limma* R package [26]. Before carrying out differential expression analysis, non-protein coding genes were excluded and lowly expressed genes were filtered using the *filterByExpr* function from R’s *edgeR* package with default parameters [27]. Expression counts were transformed using R package *limma*’s *voom* function [26]. Briefly, *voom* converts counts to log2 counts per million and assigns weights to each count observation based on its estimated variance, therefore producing data that are more suited to the assumptions of linear models. Multiple-testing correction was performed using Benjamini–Hochberg (BH) correction. Differentially expressed genes (DEGs) were defined as having an adjusted (adj.) p<0.05 and a |log(FC)|>1 [28].

### 2.3. Identification of Prognostic DEGs

Cox regression adjusted for age, tumour stage, and sex was carried out independently on DEGs to identify genes associated with OS. Prognostic genes were defined as having a 95% confidence interval that did not cross β=0 and adj. p<0.05, where *p*-values were adjusted for multiple-testing using the BH correction. We excluded features found to violate the non-proportional hazards assumption of a Cox regression with p<0.05. The resulting set of prognostic DEGs was then predicted from WSIs using attention-based multiple-instance learning (AMIL). All Cox analyses were performed using the Python *lifelines* library (version 0.27.8) [29].

### 2.4. WSI Processing

The pipeline for predicting gene expression and OS from WSIs can be divided into three broad steps, image preprocessing, feature extraction, and model training.

#### 2.4.1. Image Preprocessing

Non-overlapping patches with a size of 224×224 pixels were extracted from WSIs at a resolution of 0.5 μm per pixel. Background and blurry patches were filtered using Canny edge detection with an edge threshold of two or fewer defining the rejection criteria [30]. After standardising the brightness across individual WSIs, colour normalisation of extracted patches was carried out using the Macenko method to reduce the effect of stain variation on model training [31].

#### 2.4.2. Feature Extraction

Each processed 224×224 pixel image patch was embedded into a feature vector of size 768 using the CTransPath model [32]. CTransPath, based on the SwinTransformer architecture, leverages both the hierarchical arrangement of convolutional neural networks (CNNs) with the global self-attention mechanisms innate to transformers. Wang et al. trained CTransPath with a custom contrastive learning approach that was optimised to histopathology data by leveraging the fact that many patches both within and across WSIs share semantically similar information. This method of pretraining was shown to outperform a range of other feature extraction models, from Imagenet pre-trained CNNs to other contrastive learning-based transformers, at generating meaningful representations from histopathology data [32]. The CTransPath model and weights are publicly available at https://github.com/Xiyue-Wang/TransPath (accessed on 20 September 2023).

#### 2.4.3. AMIL Model Training

AMIL models were trained on the extracted patch features to predict selected targets [7,33]. The AMIL model was selected due to its ability to operate effectively in a weakly-supervised manner (i.e., using WSI-level labels rather than region-level annotations) and relatively efficient training time compared to other deep learning models [34]. All AMIL models were implemented in PyTorch (version 2.1.0) [35]. Min–max standardisation, carried out using the *sklearn* preprocessing library (version 1.3.1), was applied to the processed gene expression data prior to model training [36]. Similarly, external validation data were standardised using min–max scaling before deployment, and predicted scores were inverse transformed to interpret and visualise results.

Before passing the patch embeddings to the attention module, a multi-layer perceptron (MLP) with rectified linear unit (ReLU) activation was employed to embed K×768 features to output 256-dimensional vectors $h_{k}$ for each patch *k*, where K, the maximum number of tiles randomly sampled per sample at each epoch, was set to 512. The resulting feature vectors $h_{k}$ were then passed to an attention module to determine the attention score $a_{k}$ for each tile using Equation (Equation 1), where $h\in R^{256}$, $V\in R^{128\times 256}$, $w\in R^{128\times 1}$. The attention module consisted of a fully connected linear layer and a tanh activation function, followed by a second fully connected linear layer and softmax activation, resulting in an K×1 attention score vector.
(1)$$a_{k}=\frac{e^{W^{T}\tanh\left(Vh_{k}\right)}}{\sum_{j=1}^{K}e^{W^{T}\tanh\left(Vh_{j}\right)}}$$

The resulting attention score vector $a_{k}$ was multiplied by the K×256 tile-level feature vectors via Equation (Equation 2), where $h_{i}$ is the *i*-th tile’s embedding to obtain a single aggregated feature vector representing each WSI. This MIL pooling operation results in a single 1×256 feature vector $h_{sum}$ per WSI.
(2)$$h_{sum}=\sum_{K}^{i=1}a_{i}h_{i}$$

The final classification was achieved by passing the batch of $h_{sum}$ vectors through a batch normalisation layer and a dropout layer with probability set to 50%. The resulting output was finally passed to a fully connected layer with softmax activation to obtain the desired prediction. A schematic of the AMIL model architecture is shown in Figure 1c.

Models were trained for 25 epochs at each fold using the Adam optimiser and a batch size of 64 [37]. The balanced mean-squared error (Equation (Equation 3)) was used as the loss function for models trained to predict gene expression [38]. In Equation (Equation 3), *N* is the number of samples, and $w_{i}$ is the sample weight assigned to the *i*-th sample.
(3)$$\mathrm{BalancedMSE}=\frac{1}{N}\sum_{i=1}^{N}w_{i}{\left(y_{i}-{\hat{y}}_{i}\right)}^{2}$$

The Cox proportional hazards loss function was used for the survival-based model, as defined in Equation (Equation 4). All hyperparameters remained the same across the gene expression- and survival-based AMIL models. In Equation (Equation 4), *S* denotes the set of indices corresponding to observed events in the dataset, ${\hat{h}}_{\theta }$ denotes the risk score as obtained from the model’s output, *t* refers to the survival time, $x_{i}$ refers to the *i*-th sample, and $x_{j}$ refers to the *j*-th sample.
(4)$$L\left(\theta \right)=-\frac{1}{\left|S\right|}\sum_{i\in S}\left({\hat{h}}_{\theta }\left(x_{i}\right)-\log\sum_{j:t_{j}\geq t_{i}}\exp\left({\hat{h}}_{\theta }\left(x_{j}\right)\right)\right)$$

#### 2.4.4. Cross-Validation in TCGA-LUAD Dataset

Models were trained on the TCGA-LUAD dataset using 5-fold cross-validation. Deep learning models in computational pathology are known to be at risk of learning confounders in WSIs which may lead to an overestimation of model performance in cross-validation experiments [39]. To mitigate the impact of potential confounders when evaluating model performance, a tissue-source site-preserved cross-validation setup was implemented. This ensured that all samples belonging to a particular tissue-source site in the TCGA were assigned to either the training or testing fold [39]. Additionally, folds were also optimised to maintain the representative distribution of the target variables across the dataset by binning samples into 5 quantiles according to the target variable and maintaining an equal representation of quantiles in each fold. For the survival-based model, the survival time of uncensored patients was used to generate the 5 quantiles. Optimisation was carried out using the *cvxpy* (version 22.1.0.0) and *cplex* (version 1.3.1) Python packages [40,41].

### 2.5. Model Evaluation

Cross-validated Pearson’s R (i.e., the mean Pearson’s R across the test sets of the cross-validation) was used to evaluate the predictability of gene expression in the TCGA-LUAD dataset. The prognostic value of genes that could be predicted with cross-validated Pearson’s R greater than 0.4 was evaluated by aggregating predictions in the test sets of each fold and fitting a Cox regression with age, sex, and stage independently to each gene. Those genes with 95% confidence interval of the hazard ratio (HR) which did not cross 1 and p<0.05 were considered statistically significant and were selected for external validation on the CPTAC-LUAD dataset. Predicted gene expression in CPTAC-LUAD was defined as being prognostic if the 95% confidence interval of the HR did not cross 1 and p<0.05. A step-by-step representation of the study design workflow is provided in Appendix A. To facilitate comparison with previous literature aimed at evaluating OS from WSIs in TCGA-LUAD, cross-validated performance in TCGA-LUAD was evaluated according to the C-index. For each gene, a Cox regression model was fitted to the actual gene expression of the four training folds, and its performance was evaluated on the predicted expression in the corresponding test fold. This procedure was carried out for each cross-validation fold, and the mean C-index across folds and the standard deviation were reported for each gene. The survival-based AMIL model was evaluated according to the cross-validated C-index in the TCGA-LUAD dataset to enable comparison with concordance performance based on the Cox regression of predicted gene expression.

### 2.6. Patient Stratification by Real and Predicted Gene Expression

Patients in the CPTAC-LUAD dataset were stratified into low- and high-risk groups via a gene expression cut-off determined using KMeans clustering with *k* set to 2 [42]. For each gene, risk groups were generated using real and predicted gene expression separately. Performance at stratifying samples using either real or predicted gene expression was compared via Kaplan–Meier plots, log-rank *p* values, HRs, and C-indices. To calculate HRs and C-indices, the risk group was used as a covariate for Cox regression along with age, sex, and stage in order to evaluate the prognostic relevance of the WSI-derived risk group.

### 2.7. Gene Ontology Enrichment Analysis and Protein–Protein Interactions

Gene ontology (GO) enrichment analysis was carried out to identify biological processes significantly associated with the genes whose predicted expression was found to be prognostic of OS in the CPTAC-LUAD external validation dataset [43,44]. The top 20 biological processes as ranked by gene ratio were reported, where the gene ratio was defined as the proportion of genes annotated with the respective GO term relative to the total number of genes being analysed. GO enrichment was performed using the *clusterProfiler* R package (version 4.8.3) [45]. Protein–protein interactions (PPIs) were assessed with STRING (version 12.0) to understand the relatedness of genes that had predicted expression from WSIs that was prognostic of OS [46].

## 3. Results

### 3.1. Data and Participant Characteristics

Clinicopathological characteristics for the TCGA-LUAD and CPTAC-LUAD cohorts are described in Table 1. Of the 530 cases in TCGA-LUAD, 478 had WSIs available, and 517 had gene expression data available for analysis. After excluding 16 cases for failing to have the required metadata for processing, a total of 462 cases had WSI and gene expression data available. The CPTAC-LUAD cohort consisted of 110 participants, of which 99 had paired tumour WSI and gene expression data available and were therefore used for external validation.

### 3.2. Evaluating OS Directly from TCGA-LUAD WSIs

To establish a baseline and enable comparison against using predicted gene expression to evaluate OS in LUAD patients, OS was evaluated directly from the WSIs using the survival-based AMIL model. A cross-validated C-index of 0.562 (±0.067) was attained in the TCGA-LUAD dataset using the Cox proportional hazards loss function.

### 3.3. Prognostic Gene Expression in LUAD

Of the 60,660 transcripts downloaded from the GDC data repository, 9047 were brought forward for differential expression analysis after initial filtering of non-protein coding and lowly expressed genes. A total of 3307 DEGs were identified between tumour and normal tissue in the TCGA-LUAD cohort (adj. *p* < 0.05; |log(FC)|>1), of which 1291 were upregulated, and 2016 were downregulated. After applying univariate Cox regression, 318 of the 3307 DEGs failed the non-proportional hazards test and were therefore excluded from further analysis. Of the remaining genes, 239 were prognostic of OS through adjusted univariate Cox regression (95 % confidence interval that does not cross HR of 1; adj. *p* < 0.05). This set of 239 genes was subsequently used for training AMIL models in TCGA-LUAD using preserved site cross-validation to predict gene expression directly from WSIs. Results of all Cox regression analyses are provided in Appendix A.

### 3.4. Predicting Prognostic Gene Expression Directly from Whole-Slide Images

Separate AMIL models were trained to predict the expression of 239 prognostic genes directly from WSIs. Of the 239 genes, 126 (52.3%) could be predicted with cross-validated R > 0.4 and therefore had predicted expression subject to univariate Cox regression adjusted for age, sex, and stage. In total, 114 (90.5%) of the subset of 126 genes had predicted expression that was prognostic of OS. Predicting the expression of these 114 genes was externally validated in the CPTAC-LUAD dataset using the best-fold model from the cross-validation in TCGA-LUAD, as evaluated by Pearson’s R. Of the 114 genes, 112 (98.2%) were externally validated in CPTAC-LUAD with a Pearson’s R > 0.4. For illustrative purposes, correlation plots of the top nine genes as ranked by Pearson’s R are shown in Figure 2a–i. Full results of correlation analyses in TCGA-LUAD and CPTAC-LUAD are provided in Appendix A.

### 3.5. Evaluating the Prognostic Relevance of Gene Expression Predicted Directly from WSIs

The prognostic relevance of predicted gene expression was evaluated in the TCGA-LUAD dataset using the cross-validated C-index. This meant that Cox regression models were fitted to training sets and deployed on the predicted expression of the corresponding test sets, and the mean C-index across folds was determined for each gene independently. The highest mean C-index in the TCGA-LUAD dataset was attained using predicted *GAPDH* expression (C-index = 0.615 ± 0.052). Cross-validation results for the top 20 genes as ranked by the mean C-index are shown in Figure 3.

Of the 112 genes that were predictable from WSIs with R > 0.4 in the CPTAC-LUAD dataset, 36 had predicted expression that was prognostic of OS as evaluated through univariate Cox regression adjusted for age, sex, and stage. All 36 predicted gene expressions passed the non-proportional hazards test (*p* > 0.05). Log HRs and corresponding *p*-values for genes with predicted expression that was found to be prognostic in CPTAC-LUAD are shown in Figure 2.

### 3.6. Stratifying Patient-Based Gene Expression Predicted from WSIs

For each of the 36 genes that had predicted expression that was prognostic of OS, stratification according to real expression was compared with stratification according to predicted expression. Risk group was used as a covariate in Cox regression adjusting for age, sex, and stage to calculate statistics. The best-performing stratification in CPTAC-LUAD, as measured by the C-index, based on predicted gene expression was attained with *PIMREG* (C-index = 0.806). For comparison, using only the *PIMREG* expression-derived risk group as a covariate in the Cox regression yielded a C-index of 0.616 (Appendix A). On average, the C-index experienced a minimal decrease of 0.00495 when comparing stratification using real gene expression against gene expression predicted from WSIs. Notably, 14 genes (highlighted in bold in Table 2) exhibited statistically significant HRs for the risk group established from predicted gene expression (95% CIs on HR not crossing 1, and *p*-values less than 0.05), highlighting their prognostic value when adjusting for age, sex, and stage. These genes consisted of *CCNB2*, *CCNA2*, *HJURP*, *KPNA2*, *FAM83D*, *SGO1*, *MELK*, *SAPCD2*, *SLC2A1*, *PIMREG*, *TPX2*, *GAPDH*, *CDKL2*, and *ZNF540*. The C-indices, HRs, CIs, and corresponding *p*-values of the Cox regressions for the risk group derived from real and predicted gene expression of the 36 genes are shown in Table 2. Statistics for Cox regressions carried out directly on gene expression values, as well as risk group, are provided in Appendix A.

Kaplan–Meier survival analysis further validated the patient stratification based on both real and predicted gene expression. Figure 4 shows Kaplan–Meier plots for the top two genes where upregulation was associated with a survival benefit and the top two genes where downregulation correlated with improved survival, as ranked by HR (corresponding plots for all 36 genes can be found in Appendix A). C-indices, log-rank *p*-values, and HRs of risk group from adjusted Cox regression for both real and predicted expression are shown for each of the four genes. Predicted overexpression of *PIMREG* (Figure 4a) and *SLC2A1* (Figure 4b) was found to be associated with poorer OS, as evidenced by the HRs of 4.29 and 4.28, respectively. Meanwhile, predicted underexpression of *SLC15A2* (Figure 4c) and *ZNF540* (Figure 4d) were found to be associated with poorer OS.

### 3.7. GO Enrichment and PPI Network of Predictable Prognostic Genes

GO enrichment analysis of the 36 genes that had prognostic predicted expression in CPTAC-LUAD revealed a high number of biological processes involved in cell cycle regulation. The processes with the highest gene ratios included “chromosome segregation” (Gene Ratio = 0.306; adj. *p* = $2.30\times 10^{-7}$), “mitotic cell cycle phase transition” (Gene Ratio = 0.306; adj. *p* = $2.43\times 10^{-7}$), “nuclear chromosome segregation” (Gene Ratio = 0.250; adj. *p* = $2.61\times 10^{-6}$), “chromosome localization” (Gene Ratio = 0.167; adj. *p* = $3.45\times 10^{-6}$), and positive regulation of cell cycle process (Gene Ratio = 0.222; adj. *p* = $3.85\times 10^{-6}$). The top 20 enriched biological processes are shown in Figure 5a. Figure 5b shows the PPI network of the 36 genes that had predicted expression in the external dataset that was prognostic of OS. The cluster of highly connected nodes represents genes that were upregulated in patients with poor survival, while the eight genes that were downregulated in patients with poor survival have nodes with few connections to other nodes.

## 4. Discussion

LUAD is a heterogeneous disease characterised by unique molecular and histological subtypes [47]. While histopathological assessment by a trained pathologist is widely regarded as routine in cancer diagnostics, comprehensive molecular profiling remains cost-prohibitive in many clinical settings. Computational pathology promises to streamline future cancer diagnostic pipelines by enabling cost- and time-efficient evaluations of biomarkers and prognosis from readily available H&E tissue sections. Extensive research in recent years has been dedicated to evaluating genomic biomarkers such as mutations and microsatellite instability directly from WSIs using classification-based models. However, performance in end-to-end tasks, such as evaluating OS, has proved to be more challenging [8]. In this study, we show that a machine learning model aimed at predicting gene expression from LUAD WSIs can attain sufficient accuracy to have a tangible clinical application.

### 4.1. Unsupervised Contrastive Learning Pretreained Feature Extraction and AMIL Regression Can Effectively Predict Gene Expression in LUAD WSIs

In this study, we show that a machine learning framework consisting of CTransPath feature extraction and AMIL prediction can effectively learn to predict the expression of a number of genes directly from WSIs [32]. A key advantage of this method lies in its weakly supervised nature, relying only on WSI-level labels without the need for detailed annotations. Furthermore, manual annotations can introduce a source of bias in models, particularly when it is unclear which tissue type might express certain genes. Developing generalisable machine learning models in computational pathology is vital for the future clinical translation of such methods. In this study, we achieved a high degree of generalisability for models trained to predict gene expression in TCGA-LUAD when evaluated in CPTAC-LUAD. Of the 114 genes that had expression that could be predicted with an R greater than 0.4, 112 could also be predicted in CPTAC-LUAD with an R greater than 0.4.

### 4.2. Gene Expression Predicted from WSIs Can Stratify Patients with LUAD According to OS

Gene expression signatures have been widely used in research to evaluate OS in LUAD [15,16,17,18,19,20,28]. For example, Zhou et al. established a three-gene signature that achieved a C-index of 0.638 in TCGA-LUAD [28]. Through Kaplan–Meier analysis, we further confirmed that gene expression could effectively classify patients into high- and low-risk groups using KMeans clustering. For a number of genes, this relationship remained consistent in the predicted expression data, underlining the AMIL models’ ability to replicate clinically relevant trends observed in the actual gene expression data.

Chen et al. previously performed an ablation study to compare the performance of a number of deep learning models at evaluating OS in TCGA-LUAD directly from WSIs using a survival cross-entropy loss function [48]. To compare the results of this study with those obtained by Chen et al., for each gene, the predicted gene expression from the training set was used to fit a univariate Cox regression, and the model was tested on the predicted expression of patients in the corresponding test set. This procedure was carried out for each fold of the cross-validation experiments, and the mean C-index across folds was evaluated (Appendix A). In this study, we found that fitting and deploying a Cox regression on predicted *GAPDH* gene expression using cross-validation in TCGA-LUAD achieved a mean C-index of 0.615 (±0.052). Meanwhile, of the five models used by Chen et al. to evaluate OS directly from TCGA-LUAD WSIs, the graph CNN multiple-instance learning model achieved the highest C-index with 0.592 (±0.070) [48,49]. Since potential differences may exist in hyperparameters and model training protocols between this study and prior work, we evaluated the performance of a survival-based AMIL model. The survival-based model only differed from the gene expression models by its loss function. The objective of the survival-based AMIL model was to generate a risk score for each patient, offering a direct prognostic assessment based on the WSI. This model served as a benchmark to evaluate the prognostic performance of the AMIL models aimed at predicting gene expression. The cross-validated C-index of 0.562 achieved by the survival-based model in this study is substantially lower than the performance attained when using predicted gene expression to evaluate OS. This highlights that using machine learning to predict surrogate biomarkers of OS can outperform machine learning models trained to evaluate OS directly from WSIs.

### 4.3. Cell Cycle-Related Biological Processes Correlate with Tumour Histopathology

The GO enrichment analysis conducted on the 36 prognostic genes identified from the CPTAC-LUAD dataset underlines the influence of cell cycle-related biological processes on prognosis in LUAD. For instance, the enrichment of “positive regulation of cell cycle process” and “mitotic cell cycle phase transition” processes suggests that the deregulation of cell-cycle control plays an important role in tumour progression and patient outcomes, a finding that has been reported in several previous studies in the context of lung cancer [50,51,52,53,54]. Dysregulation of cell-cycle controls can lead to unchecked cellular proliferation, a hallmark of cancer, and could be captured by the model through features such as increased cell density and mitotic figures [51,55]. The predictive capacity of the AMIL model to infer gene expression from histopathological slides may be partially explained by its ability to recognise such nuclear features, which are the phenotypic manifestations of underlying genetic activity. Other enriched biological processes, such as “chromosome segregation” and “chromosome localization“ processes suggest that the spatial dynamics and accurate segregation of chromosomes may be correlated with tumour aggressiveness and patient outcome. These enriched terms may reflect the genetic instability of the tumour cells, a hallmark that has been previously described as a risk factor for LUAD [56,57]. Chromosomal instability has been previously correlated with histopathological features such as nuclear size and abnormal mitotic figures, pointing to a potential mechanism through which the AMIL model was able to learn to predict the expression of genes [58,59,60]. The prognostic capability obtained from the model’s predictions suggests that these histopathological patterns are not only reflective of tumour’s genomic state but also of its future behaviour.

### 4.4. Limitations and Future Directions

Despite the robustness of surrogate biomarker prediction for evaluating OS directly from WSIs in the external validation dataset, our study has certain limitations. Firstly, we restricted our analysis to a single cancer subtype. The predictability of gene expression from WSIs, as well as the prognostic value of genes, is likely to differ between cancer types. Secondly, despite including several hundred genes in our analysis, only 36 genes were found to have predicted expression from WSIs that was prognostic of OS. Many prognostic genes in LUAD did not have expression that could be reliably predicted from WSIs. Furthermore, many of the predictable prognostic genes showed a high degree of collinearity (Appendix A). This indicates that the final set of 36 genes tend to be differentially expressed together, limiting their potential use in constructing downstream multi-gene prognostic models. Thirdly, while GO enrichment of predictable prognostic genes reveals potential mechanisms by which the AMIL models may learn to predict gene expression, further research into the histopathological basis for such predictions is required to support these claims. Finally, our analysis has only considered LUAD patients as a single cohort without accounting for the molecular and histological heterogeneity that exists within LUADs. Each subtype of LUAD is likely to reveal a distinct set of significantly prognostic genes; however, due to currently available sample sizes with paired WSI and gene expression data, reliably interrogating the histological basis for these subtype-specific prognostic genes remains a challenge. As larger datasets become available, we anticipate that subsequent studies will be able to dissect the landscape of genes with predictable prognostic expression in LUAD with a finer-grained approach.

## 5. Conclusions

While multiple studies have reported biomarkers that are predictable from WSIs, many of these have focused on classification-based methods that fail to reach a performance that is in a clinically relevant range [61]. In this study, the successful prediction of gene expression from WSIs and the demonstrated stratification of patients based on predicted gene expression underscores the potential of deep learning models to capture clinically relevant information from histopathology data. We show that using predicted gene expression from WSIs to evaluate OS in LUAD patients performed better than predicting OS directly from the WSIs. The observed enrichment of cell cycle-related biological processes in predictable prognostic genes provides a plausible link to histopathology features, such as mitotic rates and nuclear atypia. This study sets a foundation for future research to explore the use of surrogate biomarkers evaluated from WSIs to assist in assessing patient prognosis and to extend these methodologies across other cancer types.

## Figures and Tables

**Figure 1 diagnostics-14-00462-f001:**
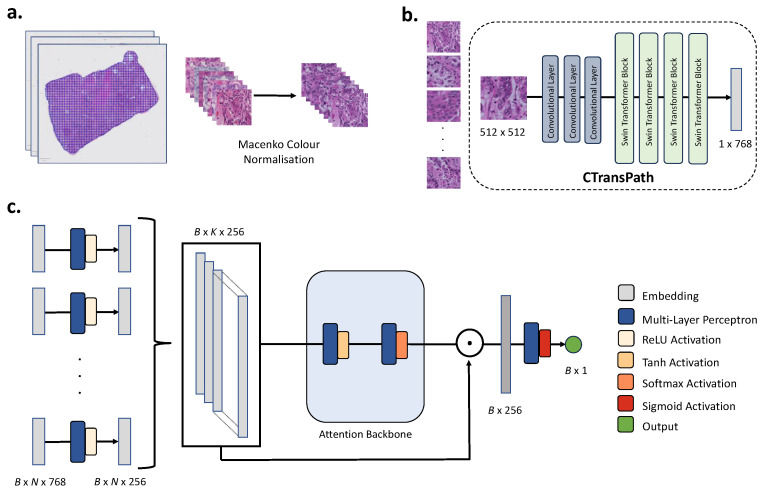
(**a**) Preprocessing of WSIs. WSIs were tiled into patches with a size of 224×224 pixels, and patches were subsequently colour normalised using the Macenko method [31]. (**b**) CTransPath feature extraction. Patches were fed to the three convolutional layers and four SwinTransformer blocks of CTransPath to extract 1×768 features per patch. (**c**) AMIL model architecture. Patch features were first passed through an MLP with ReLU activation resulting in B×K×256 feature vectors, where *B* is the batch size, and *K* is the number of patches sampled at each epoch. Attention scores were then computed for each of the inputs using two MLPs with hyperbolic tan and softmax activation, respectively. The attention scores were multiplied with each patch embedding resulting in a single 1×256 feature vector per WSI in the batch. A final MLP with sigmoid activation was used for the final prediction.

**Figure 2 diagnostics-14-00462-f002:**
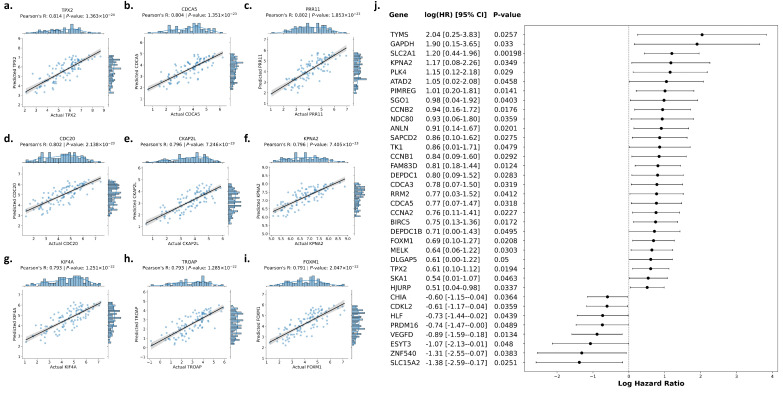
External validation on CPTAC-LUAD dataset. Correlation plots for the top 9 genes as ranked by Pearson’s R are shown. (**a**) *TPX2*. (**b**) *CDCA5*. (**c**) *PRR11*. (**d**) *CDC20*. (**e**) *CKAP2L*. (**f**) *KPNA2*. (**g**) *KIF4A*. (**h**) *TROAP*. (**i**) *FOXM1*. (**j**) Forest plot of the 36 genes that had predicted expression from WSIs in the CPTAC-LUAD dataset that was found to be prognostic of OS.

**Figure 3 diagnostics-14-00462-f003:**
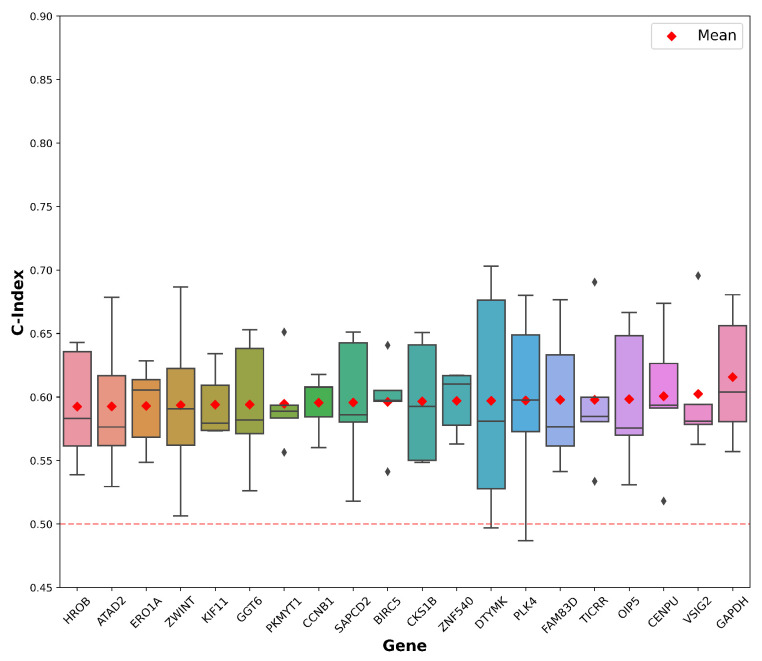
Prognostic relevance of predicted gene expression in TCGA-LUAD using cross-validated C-index. Box plots of cross-validated C-index for the top 20 genes are shown. Red diamonds represent the mean C-index across folds.

**Figure 4 diagnostics-14-00462-f004:**
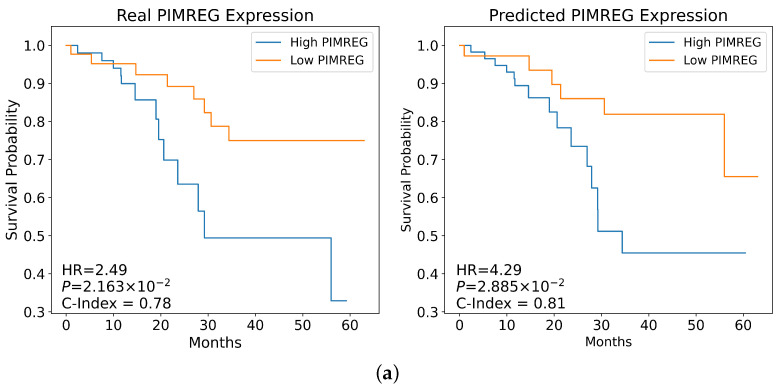
Kaplan–Meier Plots of CPTAC-LUAD patients as stratified by real and predicted gene expression. Gene expression cut-offs were determined via KMeans clustering. Hazard ratios, log-rank *p*-values, and concordance indices were determined using risk group, age, sex, and stage as covariates in a Cox regression. (**a**) Patients stratified by real and predicted *PIMREG* expression. (**b**) Patients stratified by real and predicted *SLC2A1* expression. (**c**) Patients stratified by real and predicted *SLC15A2* expression. (**d**) Patients stratified by real and predicted *ZNF540* expression. HR: hazard ratio; *p*: log-rank *p*-value; C-index: concordance index.

**Figure 5 diagnostics-14-00462-f005:**
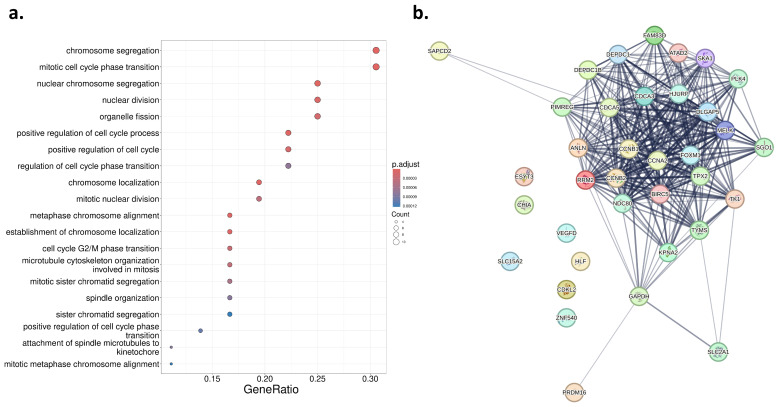
Analysis of the 36 genes that had predicted expression in CPTAC-LUAD that was prognostic of OS. (**a**) Top 20 enriched biological processes for prognostic genes predicted from WSIs in LUAD as ranked by gene ratio. Genes are ranked by the proportion of genes annotated with the respective GO term relative to the total number of genes associated with that process. Each dot represents a GO term related to biological processes, where the size of the dot indicates the count of genes involved, and the colour denotes the adjusted p-value significance levels. (**b**) PPI network of the predictable prognostic genes shows the complexity of interactions among the proteins they encode. Nodes represent individual proteins, while the thickness of the edges reflects the strength of evidence supporting each interaction.

**Table 1 diagnostics-14-00462-t001:** Clinicopathological characteristics for patients in TCGA-LUAD and CPTAC-LUAD cohorts. *p*-values denoting differences between the TCGA-LUAD and CPTAC-LUAD cohorts were derived using Fisher’s exact test for age, sex, and OS status; the Chi-square test for stage, T stage, and N stage; and the Mann–Whitney U test for follow-up months. OS: overall survival; NA: not available.

Characteristic	TCGA-LUAD				CPTAC-LUAD				*p*-Value
	N	Mean(SD)	%	% Missing	N	Mean(SD)	%	% Missing	
Age				12.54%				0.00%	$2.3\times 10^{-1}$
> 65	255	-	45.05%		45	-	40.54%		
≤ 65	240	-	42.40%		65	-	58.56%		
Sex				9.19%				0.00%	$3.4\times 10^{-3}$
Male	239	-	42.23%		72	-	64.86%		
Female	275	-	48.59%		38	-	34.24%		
Stage				9.54%				0.00%	$2.0\times 10^{-1}$
Stage I	278	-	49.11%		58	-	52.73%		
Stage II	124	-	21.91%		30	-	27.27%		
Stage III	83	-	14.66%		21	-	19.09%		
Stage IV	27	-	4.77%		1	-	0.91%		
T				9.19%				0.00%	$3.5\times 10^{-1}$
T1	169	-	29.86%		27	-	24.32%		
T2	277	-	48.94%		70	-	63.06%		
T3	46	-	8.13%		12	-	10.81%		
T4	19	-	3.36%		1	-	0.90%		
TX	3	-	0.53%		0	-	0.00%		
M				9.89%				100%	NA
M0	345	-	60.95%		NA	NA	NA		
M1	25	-	4.41%		NA	NA	NA		
MX	140	-	24.73%		NA	NA	NA		
N				9.36%				0.00%	$4.6\times 10^{-1}$
N0	331	-	58.48%		75	-	67.57%		
N1	97	-	17.14%		17	-	15.32%		
N2	73	-	12.90%		18	-	16.27%		
N3	2	-	0.35%		0	-	0.00%		
NX	10	-	1.77%		0	-	0.00%		
OS Status				9.19%				0.90%	$5.6\times 10^{-5}$
Alive	328	-	57.95%		82	-	73.87%		
Dead	186	-	32.86%		24	-	26.63%		
Follow-up Months	-	29.73 (29.52)	-	-	-	24.78 (17.75)	-	-	$2.1\times 10^{-1}$

**Table 2 diagnostics-14-00462-t002:** Stratification of CPTAC-LUAD patients into high- and low-risk groups based on real and predicted gene expression. HRs with CIs and *p*-values are reported as well as C-indices and differences in the C-index between each the Cox regression, with real and predicted expression for each gene. HRs and C-indices were computed using risk group as a covariate in Cox regression along with age, sex, and stage. Genes that had HR 95% CIs for risk group from predicted gene expression that did not cross 1 and *p*-value from the Cox regression less than 0.05 are highlighted in bold.

Gene	Real HR [CI]	Real *p*	Real C-Index	Pred. HR [CI]	Pred. *p*	Pred.C-Index	Δ C-Index
*NDC80*	6.29 [1.92, 20.67]	0.0024	0.79	2.61 [0.91, 7.49]	0.0750	0.78	−0.0036
* **CCNB2** *	4.63 [1.58, 13.59]	0.0053	0.78	3.05 [1.10, 8.47]	0.0323	0.78	−0.0004
*DEPDC1*	4.36 [1.56, 12.18]	0.0049	0.78	1.95 [0.78, 4.88]	0.1513	0.78	0.0022
* **CCNA2** *	4.36 [1.55, 12.27]	0.0052	0.80	3.25 [1.13, 9.33]	0.0285	0.77	−0.0260
*RRM2*	4.16 [1.56, 11.13]	0.0045	0.79	1.99 [0.76, 5.22]	0.1631	0.77	−0.0228
*PLK4*	3.92 [1.37, 11.23]	0.0011	0.79	2.13 [0.82, 5.53]	0.1218	0.77	−0.0197
* **HJURP** *	3.75 [1.39, 10.12]	0.0092	0.81	2.91 [1.09, 7.78]	0.0329	0.78	−0.0322
* **KPNA2** *	3.58 [1.34, 9.55]	0.0108	0.79	2.79 [1.01, 7.71]	0.0480	0.79	−0.0058
*SKA1*	3.42 [1.28, 9.10]	0.0140	0.80	2.40 [0.89, 6.42]	0.0820	0.78	−0.0242
* **FAM83D** *	3.09 [1.13, 8.43]	0.0277	0.77	3.50 [1.27, 9.68]	0.0156	0.79	0.0170
*BIRC5*	2.93 [1.14, 7.55]	0.0255	0.79	2.48 [0.94, 6.53]	0.0655	0.79	−0.0004
*FOXM1*	2.90 [1.15, 7.31]	0.0236	0.81	2.00 [0.77, 5.18]	0.1552	0.78	−0.0260
* **SGO1** *	2.86 [1.10, 7.45]	0.0310	0.78	3.05 [1.10, 8.46]	0.0325	0.78	0.0072
* **MELK** *	2.67 [1.04, 6.87]	0.0414	0.79	3.21 [1.17, 8.84]	0.0236	0.78	−0.0094
* **SAPCD2** *	2.66 [1.06, 6.66]	0.0368	0.79	3.42 [1.19, 9.87]	0.0228	0.79	−0.0076
*TYMS*	2.60 [1.02, 6.65]	0.0455	0.78	2.61 [0.98, 6.92]	0.0541	0.77	−0.0054
* **SLC2A1** *	2.56 [0.96, 6.83]	0.0595	0.78	4.28 [1.40, 13.12]	0.0109	0.80	0.0228
* **PIMREG** *	2.49 [0.96, 6.45]	0.0596	0.78	4.29 [1.50, 12.22]	0.0065	0.81	0.0210
*TK1*	2.41 [0.95, 6.12]	0.0641	0.76	1.70 [0.66, 4.39]	0.2727	0.77	0.0130
* **TPX2** *	2.40 [0.94, 6.12]	0.0678	0.79	2.97 [1.07, 8.25]	0.0367	0.78	−0.0134
*ANLN*	2.39 [0.89, 6.37]	0.0825	0.78	2.50 [0.92, 6.85]	0.0738	0.78	0.0040
*CCNB1*	2.35 [0.91, 6.06]	0.0779	0.79	2.49 [0.89, 6.98]	0.0825	0.77	−0.0134
* **GAPDH** *	2.31 [0.83, 6.43]	0.1076	0.80	3.71 [1.22, 11.30]	0.0209	0.78	−0.0130
*CDCA3*	2.24 [0.84, 6.01]	0.1081	0.79	2.40 [0.92, 6.28]	0.0751	0.78	−0.0098
*DLGAP5*	2.21 [0.86, 5.70]	0.1017	0.80	1.95 [0.74, 5.11]	0.1774	0.77	−0.0269
*CDCA5*	2.14 [0.83, 5.49]	0.1140	0.78	2.18 [0.76, 6.26]	0.1466	0.78	−0.0022
*DEPDC1B*	1.75 [0.71, 4.31]	0.2259	0.78	2.40 [0.89, 6.42]	0.0820	0.78	−0.0013
*ATAD2*	1.29 [0.52, 3.16]	0.5843	0.77	2.20 [0.84, 5.78]	0.1088	0.78	0.0107
* **CDKL2** *	0.97 [0.39, 2.38]	0.9391	0.77	0.37 [0.14, 0.96]	0.0419	0.78	0.0134
*CHIA*	0.58 [0.23, 1.44]	0.2372	0.77	0.44 [0.17, 1.15]	0.0924	0.78	0.0143
*VEGFD*	0.54 [0.22, 1.33]	0.1811	0.77	0.50 [0.19, 1.35]	0.1720	0.77	0.0027
* **ZNF540** *	0.35 [0.13, 0.91]	0.0315	0.78	0.33 [0.12, 0.93]	0.0364	0.79	0.0134
*ESYT3*	0.33 [0.13, 0.85]	0.0224	0.79	0.45 [0.15, 1.28]	0.1348	0.78	−0.0054
* **PRDM16** *	0.22 [0.08, 0.59]	0.0025	0.80	0.21 [0.07, 0.70]	0.0104	0.78	−0.0233
*HLF*	0.20 [0.07, 0.54]	0.0016	0.79	0.39 [0.14, 1.08]	0.0690	0.79	−0.0076
* **SLC15A2** *	0.15 [0.05, 0.45]	0.0008	0.82	0.22 [0.07, 0.66]	0.0069	0.80	−0.0201

Pred.: predicted HR: hazard ratio; *p*: log-rank *p*-value; C-index: concordance index.

## Data Availability

WSIs and gene expression data for TCGA-LUAD cohort are available at the GDC Portal (https://portal.gdc.cancer.gov/; accessed on 25 September 2023). WSIs for CPTAC-LUAD are available at The Cancer Imaging Archive (https://www.cancerimagingarchive.net/; accessed on 2 October 2023). Model weights for the feature extraction model, CTransPath, can be found at https://github.com/Xiyue-Wang/TransPath (accessed on 20 September 2023). Processed gene expression data and predicted gene expression data for the TCGA-LUAD and CPTAC-LUAD datasets can be found in Appendix A. Folds used for cross-validation in TCGA-LUAD can be found in Appendix A. Best-fold models trained on the TCGA-LUAD dataset that were externally validated on CPTAC-LUAD can be found in Appendix A.

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
