# Peer review of "Surrogate Biomarker Prediction from Whole-Slide Images for Evaluating Overall Survival in Lung Adenocarcinoma"

_diagnostics, 2024, doi:10.3390/diagnostics14050462_

Round 1
Reviewer 1 Report
Comments and Suggestions for Authors
This is an excellent paper on histogenetic profiling useful in prognosis of lung adenocarcinoma patients. The paper evaluates the utility of predicting the outcome of lung adenocarcinoma patients from whole slide images, using a predictive genetic profile. The methodology of the study is clearly described, and the analysis of the results is performed using advanced statistical methods.
The introduction section could be improved by adding further explanations regarding the gene expression analysis from whole-slide images.
Author Response
We would like to express our gratitude to reviewer 1 for their constructive feedback and positive comments on our manuscript. We appreciate the opportunity to improve the clarity and comprehensiveness of our work.
We have enhanced the introduction section with further explanations regarding gene expression analysis from WSIs and have included additional references to recent work in the area (PMID: 32562722 ; PMID: 32747659; PMID: 36949169). We believe that these inclusions will strengthen the introduction and provide a solid foundation for the rest of the manuscript. We thank the reviewer for their insightful comments and the editor for their consideration of our manuscript.
Reviewer 2 Report
Comments and Suggestions for Authors
In this study, the authors aimed to explore how gene expression predicted by biomarker prediction from full-screen hematoxylin and eosin (H&E) WSI images can be used to estimate overall survival (OS) in patients with pulmonary adenocarinoma (LUAD). To do this, the authors identified differentially expressed genes (DEGs) using the training set (TCGA)-LUAD. The models were externally validated on the CPTAC-LUAD dataset. Thirty-six genes were shown to have predicted expression in the external validation cohort to be prognostic of OS.
1. There are questions about the quality of the drawings: they are very small, the inscriptions are unreadable, they cannot be corrected.
2. Table 1 does not provide p-values for the two-sample comparison (TCGA)-LUAD and CPTAC-LUAD. It is clear that there are differences, but their statistical significance is unclear, in particular by gender, lymph node involvement, and survival. Add please.
3. Are the identified genes differentially expressed in the same samples or separately (some genes in some, some in others)? Is it possible to identify samples where expression of several genes from the identified list is simultaneously observed? For example, low SLC2A1 and high SLC15A2. How does the hazard ratio change then?
4. The authors did not take into account the heterogeneity of the group with adenocarcinomas: the presence of a mucinous component, the degree of differentiation, mutations in the KRAS and BRAF genes, HER2 (also known as ERBB2), fibroblast growth factor receptor 1 (FGFR1) and FGFR2, as well as the anaplastic lymphoma kinase (ALK) oncogenes ), receptor tyrosine kinase ROS1, protein tyrosine kinase Met, neurotrophic tyrosine kinase receptor type 1 (NTRK1), and RET? It is likely that for each subtype the set of prognostically significant genes in the proposed model will be different.
Author Response
We would like to express our gratitude to reviewer 2 for their constructive feedback on our manuscript. We have carefully considered each point and have made appropriate revisions to our manuscript.
Below, we address each comment individually.
1. Quality of Figures
We acknowledge the concerns raised regarding the legibility of all figures. We have regenerated all of the figures in the manuscript to ensure they are of high resolution and readable. Specifically, we increased the font size of labels and dots per inch of all figures. We hope that these adjustments make the figures in the manuscript more reader-friendly.
2. Statistical Differences between Clinicopathological Variables for TCGA-LUAD and CPTAC-LUAD Cohorts (Table 1)
This was an oversight on our behalf and we have now updated Table 1 to include the P-values for the two-sample comparison between the TCGA-LUAD and CPTAC-LUAD cohorts.
3. Differential Expression and Sample Specificity
A high degree of correlation was observed between the 36 genes with a predicted expression that was found to be prognostic of OS. This indicates that the identified genes tend to be co-expressed. We have generated a clustergram to visualise this correlation and have included it in the updated supplementary materials (Supplementary Figure 2). Specifically, the clustergram represents correlations between the predicted expression of the identified genes in the CPTAC-LUAD cohort. To address the example provided by the reviewer, we find that the predicted expression of SLC15A2 and SLC2A1 have a Pearson’s R of –0.651 and therefore suspect that including both SLC15A2 and SLC2A1 in a Cox model to stratify patients will not lead to any improvement over a single-gene model. We have revised the Limitations and Future Directions section to further highlight this high degree of correlation between the identified genes and discuss how this limits the utility of multi-gene models based on predicted expression in our study.
4. Heterogeneity of LUAD Subtypes
We acknowledge that the heterogeneity of lung adenocarcinomas may result in different sets of predictable prognostic genes across molecular subtypes. However, sample size constraints within the cohort limits our ability to identify many subtype-specific expression patterns. Specifically, due to low frequency of mutations in KRAS (31.66%), BRAF (6.55%), ERBB2 (0.87%), FGFR1 (0%), FGFR2 (0.87%), ALK (0.87%), MET (2.62%), NTRK1 (1.97%) for the TCGA-LUAD cohort, it would be infeasible to reliably evaluate differences in prognostically significant genes between these subtypes. Furthermore, specific histologic subtype information is only available for 17.1% of these cases. We have amended the Limitations and Future Directions section of the manuscript to acknowledge this limitation. Furthermore, we have highlighted that future studies may be able to dissect the landscape of genes with predictable prognostic expression in LUAD with a finer-grained approach as larger datasets with paired gene expression and WSI data become available.
We would like to thank the reviewer for their valuable comments and the editor for their consideration of our manuscript.
Round 2
Reviewer 2 Report
Comments and Suggestions for Authors
I have no further comments on the manuscript.